

# DMPNet: densely connected multi-scale pyramid networks for crowd counting

Pengfei Li, Min Zhang, Jian Wan and Ming Jiang

Computer & Software School, Hangzhou Dianzi University, Hangzhou, Zhejiang, China

## ABSTRACT

Crowd counting has been widely studied by deep learning in recent years. However, due to scale variation caused by perspective distortion, crowd counting is still a challenging task. In this paper, we propose a Densely Connected Multi-scale Pyramid Network (DMPNet) for count estimation and the generation of high-quality density maps. The key component of our network is the Multi-scale Pyramid Network (MPN), which can extract multi-scale features of the crowd effectively while keeping the resolution of the input feature map and the number of channels unchanged. To increase the information transfer between the network layer, we used dense connections to connect multiple MPNs. In addition, we also designed a novel loss function, which can help our model achieve better convergence. To evaluate our method, we conducted extensive experiments on three challenging benchmark crowd counting datasets. Experimental results show that compared with the state-of-the-art algorithms, DMPNet performs well in both parameters and results. The code is available at: https://github.com/lpfworld/DMPNet.

## INTRODUCTION

With the increase of the world population, crowd counting has been widely applied in video surveillance, crowd analysis, sporting events, and other public security services (*Chan, Liang & Vasconcelos, 2008*; *Boominathan, Kruthiventi & Babu, 2016*; *Cao et al., 2018*; *Xiong et al., 2019*). In addition, it has been extended to cell or bacterial counts in the medical field and vehicle counts in transportation field (*Xie, Noble & Zisserman, 2018*; *Hu et al., 2020*). However, crowd counting still is a challenging task due to scale variations, cluttered backgrounds, and heavy occlusion. Among these challenges scale variation is the most important research issue, as shown in Fig. 1.

Convolutional Neural Network (CNN)-based methods have made remarkable progress in crowd counting in recent years. To extract multi-scale features of crowds, researchers designed multi-column or multi-branch networks (*Sam, Surya & Babu, 2016*; *Zhang et al., 2016*; *Liu, Salzmann & Fua, 2019*; *Jiang et al., 2020*). However, most networks are limited in their ability to extract multi-scale features due to the similarity of different columns or branches (*Zhang et al., 2016*; *Sam, Surya & Babu, 2016*). In addition, multi-scale extraction modules in these networks require a lot of computation because of the complexity of the network structure (*Li, Zhang & Chen, 2018*; *Guo et al., 2019*). Our MPN also adopts

Corresponding authors
Pengfei Li, lipf@hdu.edu.cn
Min Zhang, hz_andy@163.com

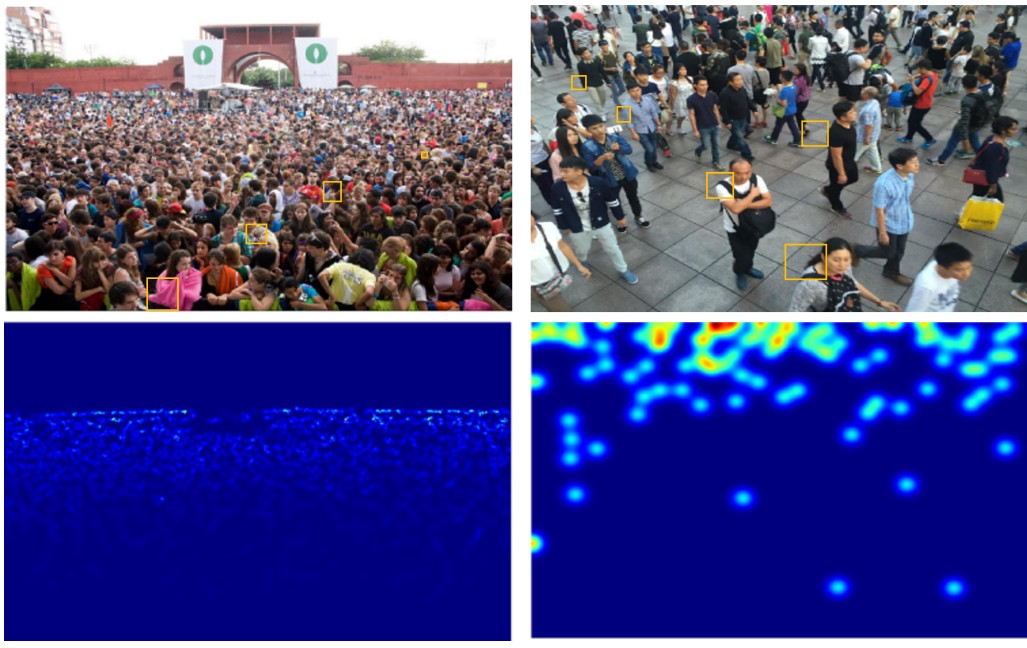

**Figure 1** **Different scales of heads exist in crowd counting datasets.** The first row shows samples of crowd images, The second row shows corresponding ground truth density maps. The samples are from ShanghaiTech Part A and Part B (*Zhang et al., 2016*).

a multi-branch structure to ensure multi-scale feature extraction, in which pyramid convolution and group convolution are used to effectively reduce parameters.

The higher resolution feature map contains finer details and the resulting density map is of higher quality, which is helpful for count estimation (*Cao et al., 2018*; *Wan & Chan, 2019*). To increase receptive fields of networks, pooling operations are adopted. However, the resolution of feature maps generated by the network become smaller, resulting in the loss of crowd image details. To keep the input and output resolutions unchanged, the encoder–decoder structure is usually utilized (*Jiang et al., 2019*; *Thanasutives et al., 2021*). The network of encoder–decoder structure uses encoder to extract input image features and combine them, and then decodes the higher-level features required by these features through a specially designed decoder. Take M-SFANet (Multi-Scale-Aware Fusion Network with Attention mechanism) (*Thanasutives et al., 2021*) for example, the encoder of M-SFANet (*Thanasutives et al., 2021*) is enhanced with ASSP (Atrous Spatial Pyramid Pooling, ASSP) (*Chen et al., 2017*), which can extract multi-scale features of the target object and fuse large context information. In order to further deal with the scale variation of the input image, they used the context module called CAN (Context Aware Network, CAN) (*Liu, Salzmann & Fua, 2019*) as the decoder. Similar to these works, we keep the input and output resolutions of MPN unchanged to ensure that the final density map generated by DMPNet contains sufficient detailed crowd information.

Different layers of neural network contain different crowd information, but with the increase of network depth, some details are gradually lost. DSNet (Dense Scale Network,

DSNet) (*Dai et al., 2021*) proposed that using dense connected networks in the field of crowd counting can effectively extract long-distance context information and maximize the retention of network layer information. We follow this operation and connect MPNs with dense connections.

Euclidean loss is the most common loss function in crowd counting SOTA methods, which is based on pixel independence (*Cao et al., 2018*; *Liu et al., 2020*; *Zhang et al., 2020*). However, texture features and pixel correlation of different regions in crowd images are different. Euclidean loss ignores the local correlation of the crowd image and does not consider the global counting error of the crowd image (*Cao et al., 2018*; *Dai et al., 2021*). Therefore, when designing the loss function, we not only consider the local density consistency of the image, but also consider the global counting loss of the image.

In this paper, we propose the densely connected Multi-scale Fig. 2. The important component Multi-scale Pyramid Network (MPN) consists of Local Pyramid Network (LPN), Global Pyramid Network (GPN), and Multi-scale Feature Fusion Network (MFFN). LPN is used to capture small heads and extract multi-scale fine-grained features, while GPN is used to capture large heads and global features. They are composed of multiple levels, and each level has filters of different sizes and depths, whose output local and global features are combined by MFFN. To maximize the flow of information between layers of the network, MPNs in the network are densely connected, with each MPN receiving as input the results of MPNs before it. To optimize the loss function, we combine Euclidean loss, density level consistency loss, and MAE loss to improve the performance of DMPNet. Experiments on three datasets (ShanghaiTech Part A and Part B, *Zhang et al., 2016*; UCF-QNRF, *Idrees et al., 2018*; UCF_CC_50, *Idrees et al., 2013*) prove the effectiveness and robustness of the proposed method.

## RELATED WORK

Generally, the existing crowd counting methods can be mainly classified into two categories: traditional methods and CNN-based methods (*Sindagi & Patel, 2017a*; *Sindagi & Patel, 2017b*; *Gao et al., 2020*). In this section, we give a brief review of crowd counting methods and explain the differences between our methods.

### Traditional methods

In early studies, detection-based methods used sliding windows to detect the target, and manually extract features of the human body or specific body parts (*Wu & Nevatia, 2007*; *Enzweiler & Gavrila, 2009*; *Felzenszwalb et al., 2010*). However, even if only heads or smaller body parts of pedestrians are detected, these methods often fail to make accurate counts of dense crowd scenes due to occlusion and illumination. To improve the performance of crowd counting, feature-based regression methods attempted to extract various features from local image blocks and generate low-level information (*Chan & Vasconcelos, 2009*; *Ryan et al., 2009*; *Ke et al., 2012*). *Idrees et al. (2013)* tried to fuse the features obtained by Fourier analysis and Scale-invariant feature transform (SIFT) interest points. However, they ignored the scale information. To overcome the problem, density estimation-based method considers the relationship between image features and data regression. *Lempitsky &*

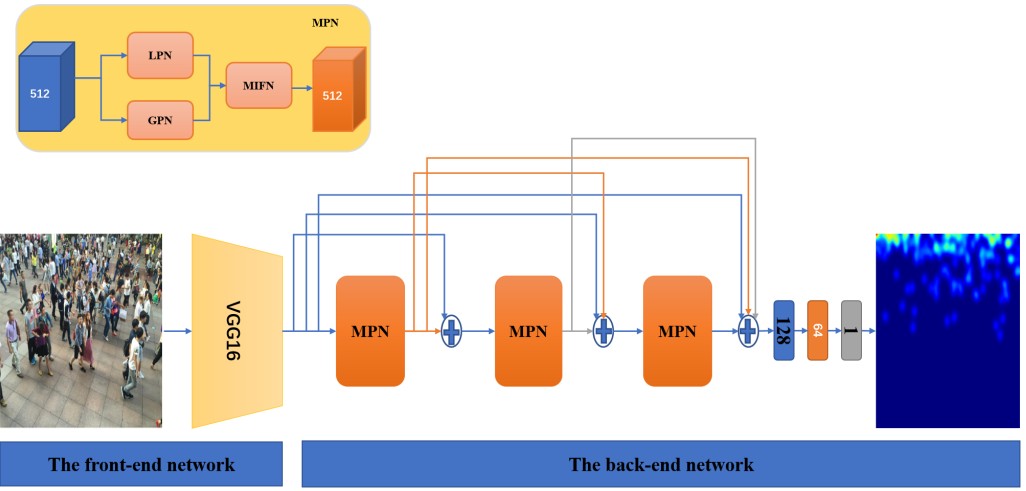

**Figure 2** **The architecture of DMPNet for crowd counting and high-quality density map.** It contains VGG16 (*Simonyan & Zisserman, 2014*) as the front-end network and three MPNs stacked by dense connections as the back-end network. MPN is composed of LPN, GPN and MFFN. It is used to extract human head features at different scales, and the resolution and channel number of input feature maps remain unchanged. The samples are from ShanghaiTech Part B (*Zhang et al., 2016*).

*Zisserman (2010)* adopted the method of extracting features in local areas and establishing linear mapping between features and density maps. *Pham et al. (2015)* tried to use random forest regression to get a nonlinear map.

## CNN-based methods

The CNN-based methods can be classified into the multi-column CNN-based methods and the single-column CNN-based methods. The multi-column CNN-based methods use multi-column networks to extract the human head features of different scales and then fuse them to generate density maps. *Zhang et al. (2016)* (Multi-Column Convolutional Neural Network, MCNN) proposed to extract features using three-column networks with convolution kernels of different sizes respectively, and fused them through 1×1 convolution. *Sam, Surya & Babu (2016)* (Switching Convolutional Neural Network, Switch-CNN) proposed to design an additional switch based on MCNN, that is to use the switch to select the most appropriate CNN column for different input images to improve the counting accuracy. Inspired by the image generation methods, *Viresh, Le & Hoai (2018)* (Iterative Crowd Counting CNN, ic-CNN) proposed a two-column networks to gradually refine the obtained low-resolution density map to high-resolution density map. *Sindagi & Patel (2017b)* (Contextual Pyramid CNN, CP-CNN) used global and local feature information to generate density maps for crowd images. *Zhang et al. (2020)* Relational Attention Network (RANet) proposed to use the stacked hourglass structure in human pose, optimized outputs from each hourglass module with local attention LSA and global attention GSA, and then fused the two features with a relational module. *Zhu et al. (2019)* (Multi-Scale Fusion Network with Attention mechanism, SFANet) proposed a dual path multi-scale fusion network architecture with attention mechanism, which contains a VGG

as the front-end feature map extractor and a dual path multi-scale fusion networks as the back-end to generate density map. *Jiang et al. (2020)* (Attention Scaling Network, ASNet) proposed to use different columns to generate density maps and scale factors, then multiply them by the mask of the region of interest to output multiple attention-based density maps, and add the density maps to obtain a high-quality density map. These methods have a strong ability in extracting multi-scale features and improving the performance of crowd counting. However, they also have some disadvantages: these networks usually have a lot of parameters, and the similarity of networks with different columns results in limited feature extraction ability. In addition, training multiple CNNs at the same time will lead to slower training speed (*Li, Zhang & Chen, 2018*; *Cao et al., 2018*; *Jiang et al., 2020*).

The single-column CNN methods try to use the multi-branch structure for optimization, which can extract multi-scale information and effectively reduce parameters (*Li, Zhang & Chen, 2018*; *Cao et al., 2018*; *Liu, Salzmann & Fua, 2019*). *Li, Zhang & Chen (2018)* (Congested Scene Recognition Network, CSRNet) proposed the network structure of the front and back end, in which the front-end network adopts VGG16 (*Simonyan & Zisserman, 2014*), and the back-end network uses dilated convolution to increase the receptive field and extract multi-scale features. *Cao et al. (2018)* (Scale Aggregation Network, SANet) proposed to extract multi-scale features by using convolution containing multiple levels, and the convolution kernel of each level is different in size. At the back end of SANet, the resolution of the feature map is restored to the size of the input image by deconvolution, and the final density map is obtained. *Liu, Salzmann & Fua (2019)* (Context Aware Network, CAN) proposed a pooling pyramid network to extract multi-scale features and adaptively assign weights to crowd regions of different scales in images. *Shi et al. (2019)* (Perspective-Aware CNN, PACNN) proposed a perspective-aware network, which can integrate the perspective information into density regression to provide additional knowledge of scale variations in images. *Miao et al. (2020)* (Shallow feature based Dense Attention Network, SDANet) proposed to reduce the influence of background by introducing an attentional model based on shallow features, and to capture multi-scale information through dense connections of hierarchical features. *Thanasutives et al., (2021)* (M-SFANet) proposed to use ASPP (*Chen et al., 2017*) containing parallel atrous convolutional layers with different sampling rates to enhance the network, which can extract multi-scale features of the target object and incorporate larger context. *Jiang et al., (2019)* (Trellis Encoder-Decoder Network, TEDNet) proposed to build multiple decoding paths in different coding stages to aggregate features of different layers. *Ma et al. (2019)* (Bayesian Loss, BL) regarded crowd counting as a probability problem, the predicted density map is a probability map, each point represents the probability of existence at the point, and each point of the density map is regarded as the sample observation value.

Our DMPNet is a single-column network with multi-branch, similar to some works (*Cao et al., 2018*; *Liu, Salzmann & Fua, 2019*; *Dai et al., 2021*). We differ them from three aspects: (1) Each branch of our convolution kernel is not only different in size, but also different in the number of channels, which improves the ability of feature extraction of similar networks. (2) We use group convolution to process convolution kernels of different sizes, effectively reducing network parameters, and the calculation process is similar to

Google MixNet (Mixed Depthwise Convolutional Network, MixNet) (*Tan & Le, 2019*). (3) Our DMPNet is an end-to-end architecture, without adding extra perspective maps or attention maps (*Shi et al., 2018*).

## METHODS

The basic idea of our approach is to implement an end-to-end network that can capture multi-scale features and generate a high-quality density map, to achieve accurate crowd estimation. In this section, we first introduce our proposed DMPNet architecture, then present our loss function.

### DMPNet architecture

Similar to CSRNet (*Li, Zhang & Chen, 2018*), our DMPNet architecture includes a front-end network and a back-end network (see Fig. 2). In the front-end network, the first ten layers with three pooling layers of VGG16 are used to extract features from crowd images. Several works have proved that VGG16 achieves a trade-off between accuracy and computation, and is suitable for crowd counting (*Cao et al., 2018*; *Viresh, Le & Hoai, 2018*; *Wan & Chan, 2019*). In the back-end network, MPNs that can extract multi-scale features are connected in a dense way to improve information flow between layers. The integration between the different layers in the network can also be further retained multi-scale features (*Huang et al., 2016*; *Miao et al., 2020*; *Amaranageswarao, Deivalakshmi & Ko, 2020*). In ablation experiments, we demonstrated the effectiveness of dense connections.

### Multi-scale pyramid network (MPN)

MPN consists of three parts: Local Pyramid Network (LPN), Global Pyramid Network (GPN), and Multi-scale Feature Fusion Network (MFFN), illustrated in Fig. 2. The design principle of MPN is to keep the resolution and channel number of input and output features unchanged, and effectively extract multi-scale features.

### Pyramid convolution and group convolution

Pyramid convolution has been applied to image segmentation, image classification and other fields, and achieved remarkable results (*Lin et al., 2017*; *Duta et al., 2020*; *Wang et al., 2020*; *Richardson et al., 2020*). Inspired by this, we propose to apply pyramid convolution to crowd counting. Compared to standard convolution, pyramid convolution is composed of convolution kernels of different sizes and depths in N level, without increasing computational cost and complexity, illustrated in Fig. 3.

Each level of pyramid convolution is computed with all input features. To use different depths of the kernels at each level of pyramid convolution, we do this using group convolution. The input features are divided into four groups, and the convolution kernels are applied separately for each input group, illustrated in Fig. 4.

We compare the parameters of standard convolution and group convolution. (1) Standard convolution contains a single type of convolution kernel (with height K, width K), and the depth is equal to the number of channels of input features $C_1$. $C_2$ such convolution kernels and input features (with height H, width W) are calculated to get output features

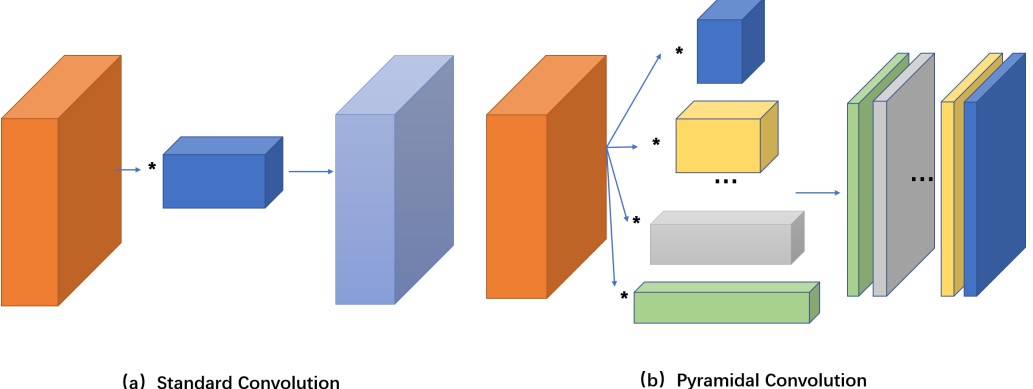

(a) Standard Convolution                    (b) Pyramidal Convolution

**Figure 3    Compare the calculation process of standard convolution and pyramid convolution.** In pyramid convolution, the input feature map is calculated with convolution kernels of different sizes, and then the obtained feature map is connected by channel as the output feature map. The size of convolution kernel is decreasing, and the depth of convolution kernel is increasing.

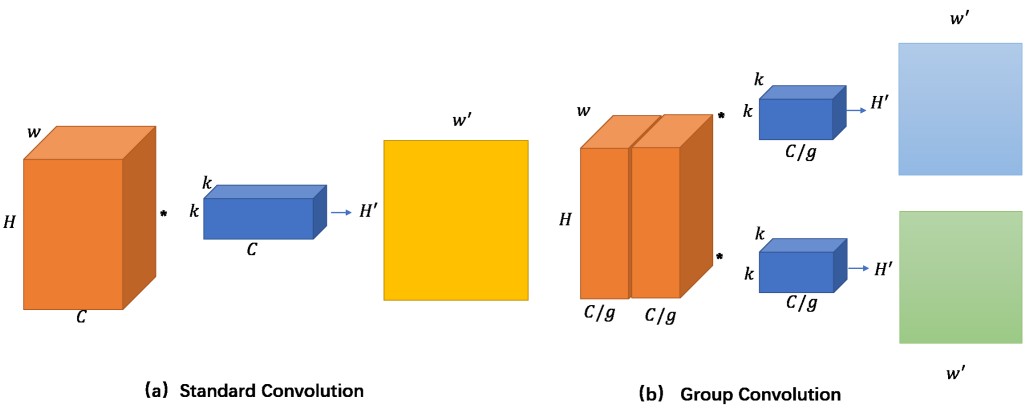

(a) Standard Convolution                    (b) Group Convolution

**Figure 4    Compare the calculation process of standard convolution and group convolution.** In grouping convolution, the input feature map is divided into N groups, and the convolution kernel is also divided into N groups accordingly. The calculation is carried out in the corresponding group. Each group will generate a feature map, and a total of N feature maps are generated.

(with height $H'$, width $W'$). Therefore, the parameter number of standard convolution is $k^2 C_1 C_2$. (2) Group Convolution divides the input feature map (with height H, width W) into $g$ groups, the depth is equal to the number of channels of input features $C_1$, and then performs convolution calculation within each group. The convolution kernels (with height K, width K, and the number of channels $C_2$) are also divided into corresponding $g$ groups. Each group of convolution generates feature maps (with height W', width H', and the number of channels $C_2/g$). Therefore, the parameter number of group convolution is $k^2 * \left(\frac{C_1}{g}\right) * \left(\frac{C_2}{g}\right) * g = k^2 C_1 C_2/g$. The width and height of the output depend on the convolution step size, and these two values are not considered here. The above calculation results prove that group convolution can generate feature maps with fewer parameters.

The more feature maps, the more information that can be encoded for the crowd counting network.

## LPN, GPN, and MFFN

Based on the ability of pyramid convolution and group convolution, we design LPN and MPN to extract local and global features of crowd images, and use MFFN to combine the two, as shown in Fig. 5.

(a) LPN is mainly used for fine-grained feature extraction. Detailed information is shown in Fig. 5A. First, we use 1x1 convolution to reduce the channel of $F_I$ to 512. Then, four-level pyramid convolution with different convolution kernels sizes ($9 \times 9$, $7 \times 7$, $5 \times 5$, and $3 \times 3$) is used to extract multi-scale features. The corresponding channel number is 32,64,128,256, and the group convolution size is 16,8,4,1. Finally, we use 1x1 convolution to increase the channel numbers of the four-level features to 512, and the output feature $F_L$ is obtained. All convolution operations are followed by BN and ReLU.

(b) GPN is mainly used for coarse-grained feature extraction. Detailed information is shown in Fig. 5B. The intermediate processing of GPN and LPN is the same, but the difference is that the input feature $F_I$ first goes through a layer of 9x9 adaptive average pool to ensure that complete global information can be obtained. In addition, to restore the resolution of the output feature map, we use bilinear interpolation for up-sampling to obtain the final output $F_G$.

(c) MFFN is mainly used for global and local feature fusion (fine-grain and coarse-grain features). Detailed information is shown in Fig. 5C. First, the output of LPN and GPN is combined into the features with 1024 channels as the input of MFFN. Then, through a layer of $3 \times 3$ convolution output the features of 256 channels. Finally, we use $1 \times 1$ convolution to restore the channel numbers to 512 and obtain feature $F_o$.

## Loss function

Euclidean loss is the most common loss function in crowd counting. It evaluates the difference between the ground truth and the estimated density map based on pixel independence, without considering the local density correlation of images. However, the local features of the crowd are generally consistent. In addition, Euclidean loss does not consider the counting error of the image (*Cao et al., 2018*; *Dai et al., 2021*). Therefore, we combine density-level consistency loss and MAE loss with Euclidean loss in the loss function.

### Euclidean loss

Euclidean loss can estimate the pixel-level error between the estimated density map and the ground truth. It is the most common loss function in crowd counting. The Euclidean loss function can be defined as follow:

$$L_E = \frac{1}{N} \sum_{i=1}^{N} ||\mathrm{F}(X_i; \theta) - F_i||_2^2$$

where N is the size of training batch, $\theta$ is the variable parameters of DMPNet. $X_i$ is the input image, $F_i$ represent the ground truth, and $\mathrm{F}(X_i; \theta)$ is the output of DMPNet.

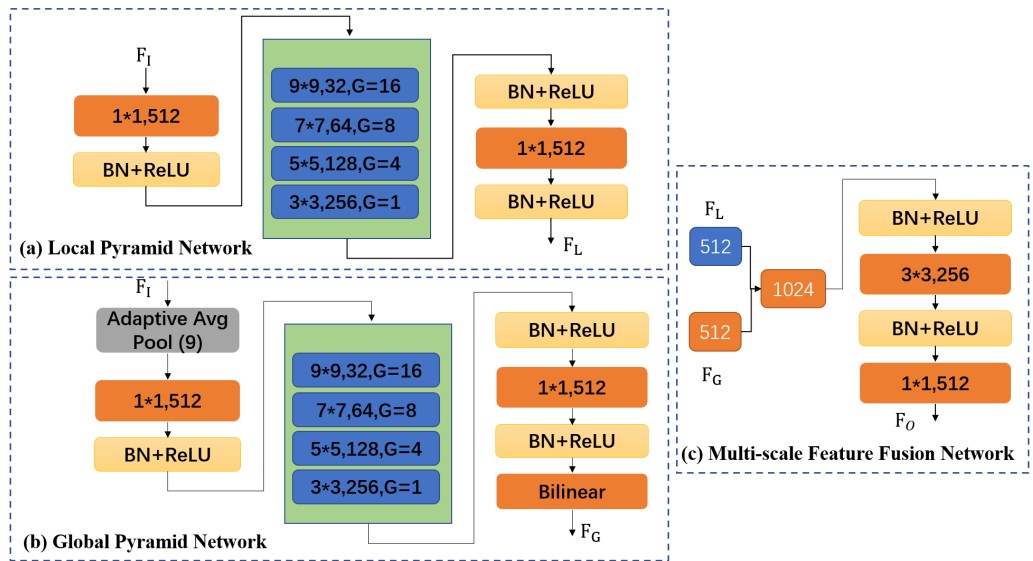

**Figure 5** **Three main components of multi-scale pyramid network.** $F_I$ is the input features of LPN and GPN. $F_L$ and $F_G$ are output features of LPN and GPN, respectively. $F_O$ is output features of MFFN.

### Density level consistency loss

Due to the imbalance of crowd distribution, the density map has a local correlation, and the density level of different sub-regions is not the same. Therefore, the density map generated by the model should be consistent with the ground truth (*Wan & Chan, 2019*; *Jiang et al., 2020*). Referring to the setting of reference (*Dai et al., 2021*), we divide the density map into sub-regions of different sizes and formed pool representations. Three outputs of different sizes are used ($1 \times 1$, $2 \times 2$, $4 \times 4$), with $1 \times 1$ representing the global density level of the density map and the other two representing the density level of different local sizes in the density map. The density level consistency loss can be defined as follow:

$$L_D = \frac{1}{N} \sum_{i=1}^{N} \sum_{j=1}^{S} \frac{1}{k_j^2} ||P_{ave}\left(F(X_i;\theta),k_j\right) - P_{ave}(D_i^{GT},k_j)||_1$$

where $S$ represents the number of scale levels, $P_{ave}$ is the average pooling operation, and $k_j$ represents the specified output size of average pooling.

### MAE loss

Mean absolute error (MAE) loss can estimate the real count and the estimated count. The MAE loss can be defined as follow:

$$L_A = \frac{1}{N} \sum_{i=1}^{N} |C(I_i) - C^{GT}(I_i')|$$

where $I_i$ and $I_i'$ represent the density map generated by DMPNet and the real density map of $X_i$ separately. $C$ represents the sum of all pixels. $C(I_i)$ and $C^{GT}(I_i')$ represent the estimated count and the real count of $X_i$ separately.

**Table 1** **The setups for different datasets.** Parameter settings for density maps generated from different datasets.

| Datasets | Parameter settings |
|---|---|
| ShanghaiTech Part_A (*Zhang et al., 2016*) | $\sigma_i=4$ |
| ShanghaiTech Part_B (*Zhang et al., 2016*) | $\sigma_i=15$ |
| UCF_QRNF (*Idrees et al., 2018*) | Geometry-adaptive kernels |
| UCF_CC_50 (*Idrees et al., 2013*) | Geometry-adaptive kernels |

### The final loss

The final loss consists of $L_s, L_c$, and $L_E$. $\alpha$ and $\beta$ are weighting factors of $L_s$ and $L_c$. According to our experiments, they are set as 10-4 and 10-3, separately.

$$L(\Theta) = L_E + a L_D + \beta L_A.$$

## EXPERIMENTAL AND DISCUSSION

### Training methods

#### Ground truth generation

The ground truth density map can represent the image containing N people. Following the methods in (*Zhang et al., 2016*; *Liu, Salzmann & Fua, 2019*; *Jiang et al., 2020*), We convolve $\delta(x - x_i)$ with a Gaussian kernel $G_{\sigma_i}(x)$ (which is normalized to 1) with parameter $\sigma_i$ to blur each head annotation. The ground truth density map can be defined as follow:

$$F(x) = \sum_{i=1}^{N} \delta(x - x_i) * G_{\sigma_i}(x) \text{ with } \sigma_i = \beta \; \bar{d}^i$$

where $x_i$ represents the position of pixel, $\bar{d}^i$ is the average distance of $k$ nearest neighbors, $\beta$ is a constant. We set $k = 3$ and $\beta = 0.3$. $\sigma_i$ is standard deviation, the setups are shown in Table 1.

#### Training details

Our DMPNet is implemented based on the PyTorch framework. It consists of a front-end network with the first 10 layers of VGG16 (*Simonyan & Zisserman, 2014*) and a back-end network with three densely connected MPNs. The training batch size is 1, optimized by Adam (*Kingma & Ba, 2014*), and the learning rate is 5e−6 and the weight decay of 5e−4. Random Gaussian initialization with 0.01 standard deviation is used. Besides, we perform data enhancement on the image, and the enhancement principle followed CSRNet (*Li, Zhang & Chen, 2018*). Considering the illumination changes, we carry out gamma transform and gray transform on images, and the transformation principle follows DSNet (*Dai et al., 2021*).

### Evaluation metrics

Mean Absolute Error (MAE) and Root Mean Squared Error (RMSE) can evaluate the performance of crowd counting (*Wan & Chan, 2019*; *Jiang et al., 2020*). MAE and RMSE represent the accuracy and robustness of the network respectively, and they can be defined as follows:

$$MAE = \frac{1}{N}|D_i - D_i^{GT}|$$
$$RMSE = \sqrt{\frac{1}{N}\sum_{i=1}^{N}(D_i - D_i^{GT})^2}$$

where N is the number of test images. $D_i$ and $D_i^{GT}$ represent the actual and estimated numbers of people in the i-th image respectively.

## Datasets

We evaluate DMPNet on three benchmark crowd counting datasets: ShanghaiTech (*Zhang et al., 2016*), UCF-QNRF (*Idrees et al., 2018*), UCF CC 50 (*Idrees et al., 2013*). (1) ShanghaiTech: It includes Part A and Part B, with a total 1,198 images and 330,165 annotations. Part A contains 300 training images and 182 testing images for congested crowd scenes, counting from 33 to 3,139. Part B contains 400 training images and 316 testing images, for sparse crowd scenes, counting from 9 to 578. (2) UCF-QNRF: It is the largest and most recently released dataset on crowd counting with 1,535 dense crowd images from various websites, counting from 49 to 12,865. (3) UCF CC 50: It contains 50 images with 63,974 annotations, counting from 94 to 4,543. The average number of people in the image is 1,280.

## Comparison with State-of-the-Art

We evaluate and compare our DMPNet and SOTA methods on three challenging crowd counting datasets. The experimental results are shown in Table 2. As you can see, our DMPNet is in the top two in multiple comparisons. (1) On ShanghaiTech part A (*Zhang et al., 2016*), MAE of our model is 98.3, which is the second best result. RMSE is 63.7, 7.2% higher than that of the optimal model RANet (*Zhu et al., 2019*). On ShanghaiTech Part B (*Zhang et al., 2016*), MAE and RMSE are 13.4% and 15.6% higher than DSNet (*Dai et al., 2021*) and SDANet (*Miao et al., 2020*), respectively. The images of Part A are from the Internet with highly congested scenes. The images of Part B come from streets captured by fixed cameras with relatively sparse crowd scenes. It indicates that our DMPNet can perform well both congested and sparse crowd scenes. (2) On UCF_QNRF (*Idrees et al., 2018*), although we do not reach the best, we still have a good performance. MAE and RMSE are 98.7 and 179.8, respectively, 15.3% and 18.9% higher than M-SFANet (*Thanasutives et al., 2021*). UCF_QNRF has lots of different scenes, in which the viewpoint and lighting variations are more diverse. In addition, due to the great change of crowd density, the perspective distortion of the head is more serious. Our model can handle this data well, which proves that our model has a certain adaptability to multiple scenes. In the face of crowd images close to real high-density scenes in UCF_QNRF, DMPNet can produce more accurate counting. (3) On UCF_CC_50 (*Idrees et al., 2013*), 5-fold cross-validation is used to evaluate our DMPNet and achieve the second-best results of MAE and RMSE, 24.7% and 25.3% higher than M-SFANet (*Thanasutives et al., 2021*) and DSNet (*Dai et al., 2021*) respectively. UCF_CC_50 is a challenging dataset with few samples and low image resolution. The results of this data demonstrate that we can also achieve high results on small datasets.

The visualization results of our DMPNet are shown in Fig. 6, and the quality of density maps generated by DMPNet and SOTA methods is compared on ShanghaiTech Part A and

**Table 2  Comparisons of our DMPNet with SOTA methods.** The empirical comparison of three mainstream datasets shows that our method is more effective on MAE and MSE. We have bolded the best two results from each dataset.

| Methods | ShanghaiTech Part A | | ShanghaiTech Part B | | UCF_QNRF | | UCF_CC_50 | |
|---|---|---|---|---|---|---|---|---|
| | MAE | RMSE | MAE | RMSE | MAE | RMSE | MAE | RMSE |
| MCNN | 110.2 | 173.2 | 26.4 | 41.3 | 277.0 | 426.0 | 377.6 | 509.1 |
| Switch-CNN | 90.4 | 135.0 | 21.6 | 33.4 | 228.0 | 445.0 | 318.1 | 439.2 |
| CP-CNN | 73.6 | 106.4 | 20.1 | 30.1 | – | – | 295.8 | 320.9 |
| ic-CNN | 68.5 | 116.2 | 10.7 | 16.0 | – | – | 260.9 | 365.5 |
| CSRNet | 68.2 | 115.0 | 10.6 | 16.0 | – | – | 266.1 | 397.5 |
| SANet | 67.0 | 104.5 | 8.4 | 13.6 | – | – | 258.4 | 334.9 |
| BL | 62.8 | 101.8 | 7.7 | 12.7 | **88.7** | **154.8** | 229.3 | 308.2 |
| RANet | **59.4** | 102.0 | 7.9 | 12.9 | 111 | 190 | 239.8 | 319.4 |
| SDANet | 63.6 | 101.8 | 7.8 | **10.2** | – | – | 227.6 | 316.4 |
| SFANet | **59.8** | 99.3 | 6.9 | 10.9 | 100.8 | 174.5 | 219.6 | 316.2 |
| PACNN | 66.3 | 106.4 | 8.9 | 13.5 | – | – | 241.7 | 320.7 |
| TEDNet | 64.2 | 109.1 | 8.2 | 12.8 | 113 | 188 | 249.4 | 354.5 |
| DSNet | 61.7 | 102.6 | **6.7** | **10.5** | 91.4 | 160.4 | **183.3** | **240.6** |
| M-SFANet | 59.69 | **95.66** | **6.76** | 11.89 | **85.60** | **151.23** | 162.33 | 276.76 |
| DMPNet | 63.7 | **98.3** | 7.6 | 11.8 | 98.7 | 179.8 | 202.4 | 301.5 |

Part B datasets (*Zhang et al., 2016*) are shown in Fig. 7. The comparison of visualization results and counting results shows that DMPNet can extract different types of crowd image information, and the density map is closer to the ground truth and higher in counting accuracy than MCNN (*Zhang et al., 2016*) and CSRNet (*Li, Zhang & Chen, 2018*). Our DMPNet has well solved the problems of crowd occlusion, perspective distortion, and scale variations.

## Ablation experiments

In this subsection, we perform several ablation experiments including Multi-scale Pyramid Network (*i.e.,* LPN, GPN, and LPN+GPN), connected network (*i.e.,* dense connection and without dense connection), and loss function. Following the previous works (*Li, Zhang & Chen, 2018*; *Jiang et al., 2020*; *Zhang et al., 2020*), ablation experiments are conducted on ShanghaiTech Part A (*Zhang et al., 2016*).

### *Effect of LPN and GPN*

To verify the effects of LPN and GPN, we adjust the network structure with three different combinations. The results of LPN and GPN are summarized in Table 3. In comparison, LPN achieves better results than GPN, with MAE and MSE lower 4.4% and 7.1%, respectively. When the two networks are used together, the results are further reduced by 5.4% and 6.7% relative to LPN. The results show that the proposed multi-scale extraction module is effective in capturing coarse-grained and fine-grained scales.

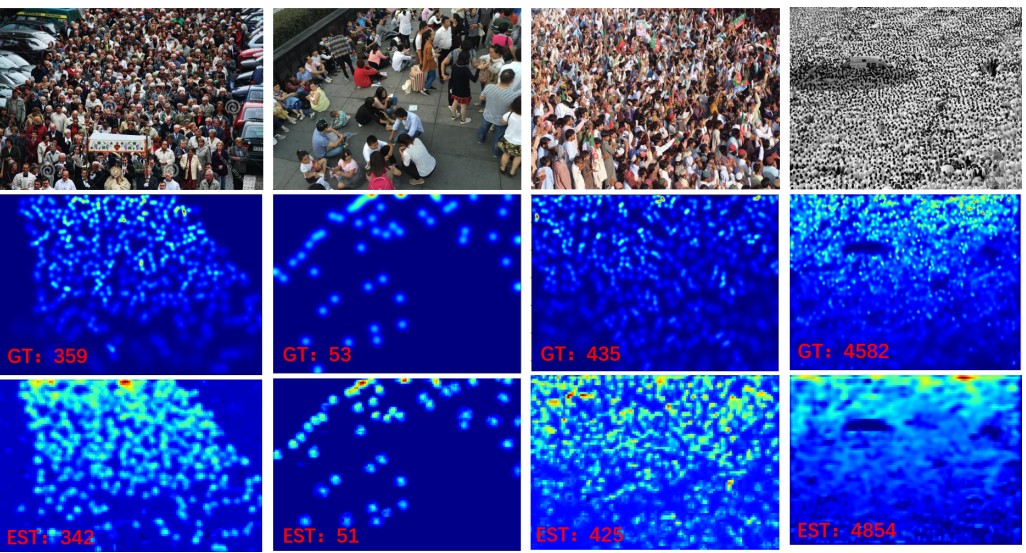

**Figure 6** **The visualization results and the corresponding counting results of our DMPNet.** The first row illustrates different test images from left to right: ShanghaiTech Part A (*Zhang et al., 2016*), ShanghaiTech Part B (*Zhang et al., 2016*), UCF-QNRF (*Idrees et al., 2018*), and UCF_CC_50 (*Idrees et al., 2013*). The second and third lines are the ground truth map and the estimated density map generated by DMP-Net, respectively.

### Effect of Dense connection

To verify the effects of dense connections, we compare two structures, one with dense connections and the other without dense connections, and the results are shown in Table 4. Results are significantly better when dense connections are used, with MAE and MSE decreasing by 6.8% and 9.5%, respectively. This indicates that dense connection effectively prevents feature loss, increases information flow between different network layers, further enlarges scale diversity, and makes the feature more effective.

### Effect of loss function

To verify the effect of different loss function combinations, we design four different combinations, and the results are shown in Table 5. MSE Loss, as the most common loss function in crowd counting, still plays a major role. However, after density level consistency loss and MAE loss are added, the effect is improved to a certain extent. When both are used, MAE and MSE decrease by 9.0% and 9.3%, respectively, indicating that the combination of density level consistency loss and MAE loss can help the model to better converge and improve the counting performance.

### Effect of the number of MPN

In order to verify the influence of the number of MPNs on the results, the number of MPNs is gradually increased and dense connections are used in different structures. The results are shown in Table 6. When the number of N is not greater than 3, the result of crowd counting is better as the number of MPN increases. When $N = 3$, MAE and RMSE are 63.7 and 98.3, respectively. When $N = 4$, the results were 64.4 and 97.7, with no significant improvement.

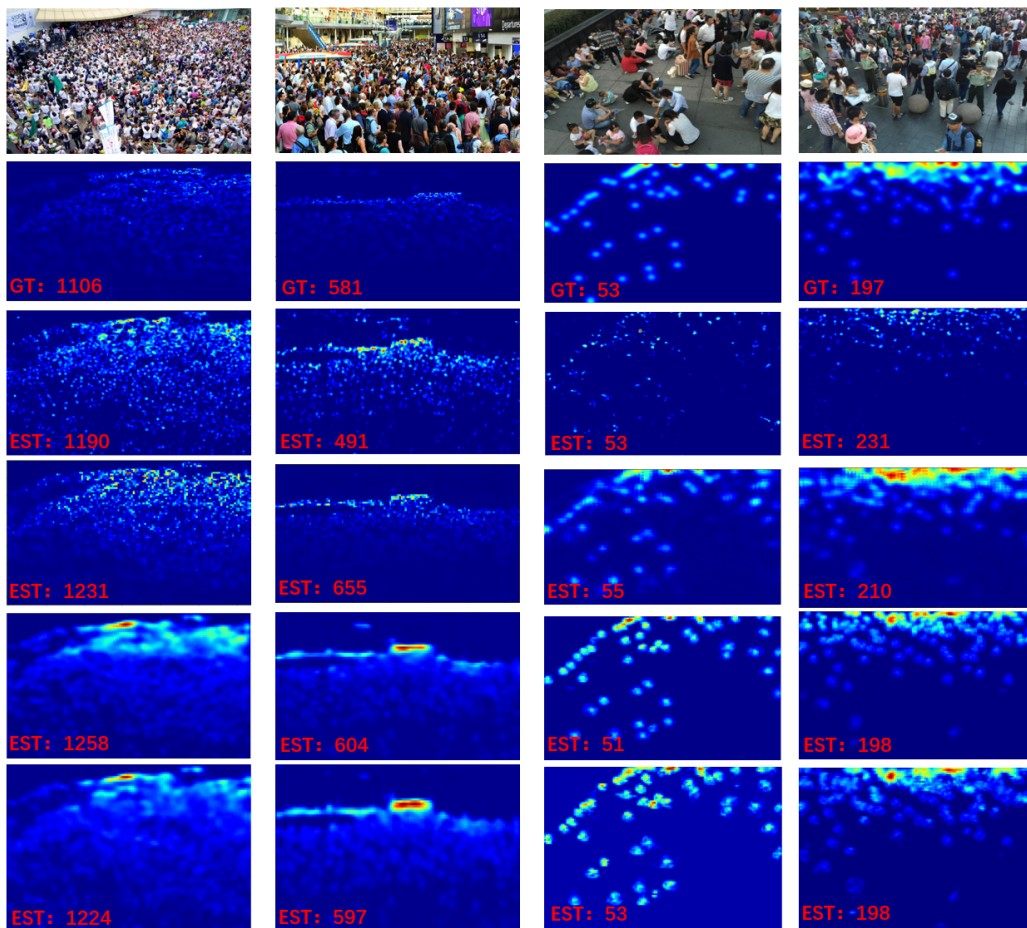

**Figure 7  Comparison of density maps generated by different SOTA methods on ShanghaiTech Part A and Part B dataset (*Zhang et al., 2016*).** The six rows show: (1) The test images; (2) the ground truth; (3) density maps produced by MCNN (*Zhang et al., 2016*); (4) density maps produced by CSRNet (*Li, Zhang & Chen, 2018*); (5) density maps produced by DSNet (*Dai et al., 2021*); (6) density maps produced by our DMPNet.

**Table 3  The estimation errors of LPN and GPN are compared on ShanghaiTech Part A (*Zhang et al., 2016*).** In the following training, MFFN is used.

| Methods | MAE | RMSE |
| --- | --- | --- |
| w/ LPN, w/o GPN | 67.3 | 105.4 |
| w/ GPN, w/o LPN | 70.4 | 113.5 |
| w/ (LPN+GPN) | 63.7 | 98.3 |

**Table 4** The estimation errors of dense connections are compared on ShanghaiTech Part A (*Zhang et al., 2016*). In the following training, we used three MPNs.

| Method | MAE | RMSE |
|---|---|---|
| w/o Dense connection | 68.4 | 108.7 |
| w/ Dense connection | 63.7 | 98.3 |

**Table 5** The estimation errors of different loss function combinations are compared on ShanghaiTech Part A (*Zhang et al., 2016*).

| Method | MAE | RMSE |
|---|---|---|
| $L_E$ | 70.0 | 108.4 |
| $L_E + L_D$ | 67.3 | 105.8 |
| $L_E + L_A$ | 69.6 | 107.6 |
| $L_E + L_D + L_A$ | 63.7 | 98.3 |

**Table 6** The estimation errors of different MPN numbers are compared on ShanghaiTech Part A (*Zhang et al., 2016*). MPN(n) represents that the network contains n MPNs.

| Method | MAE | RMSE |
|---|---|---|
| $MPN(1)$ | 71.0 | 111.3 |
| $MPN(2)$ | 66.2 | 103.4 |
| $MPN(3)$ | 63.7 | 98.3 |
| $MPN(4)$ | 64.4 | 97.7 |

In DMPNet, we use dense connection, so there is no need to set too many MPN numbers, which will cause the increase of parameters and the redundancy of calculation.

## CONCLUSION

In this paper, we proposed a novel end-to-end model called DMPNet for accurate crowd counting and high-quality density map generation. The front-end network of DMPNet is VGG16, and the back-end network is stacked by three densely connected MPNs. As an important component module of DMPNet, MPN can effectively extract multi-scale features while keeping the input and output resolution unchanged. The ability of the network is further enhanced by densely connecting multiple MPNs. In addition, we combined Euclidean loss with density level consistency loss and MAE loss to further improve the effect of the model. Experimental results on three challenging datasets validate the adaptability and robustness of our method in different crowd scenes. Although we deal with scale variation well, we did not eliminate background noise in the crowd density map, which will affect the counting accuracy to some extent. In future work, we will introduce attention mechanism to deal with background noise.

### Funding

This work is supported by the Zhejiang Provincial Technical Plan Project (No. 2020C03105, 2021C01129), and the Xiaoshan District Science and Technology Plan Project (No. 2020102). The funders had no role in study design, data collection and analysis, decision to publish, or preparation of the manuscript.

### Grant Disclosures

The following grant information was disclosed by the authors:
Zhejiang Provincial Technical Plan Project: No. 2020C03105, 2021C01129.
Xiaoshan District Science and Technology Plan Project: No. 2020102.

### Competing Interests

The authors declare there are no competing interests.

### Author Contributions

- Pengfei Li conceived and designed the experiments, performed the experiments, analyzed the data, performed the computation work, prepared figures and/or tables, authored or reviewed drafts of the paper, and approved the final draft.
- Min Zhang conceived and designed the experiments, performed the experiments, analyzed the data, authored or reviewed drafts of the paper, and approved the final draft.
- Jian Wan performed the experiments, analyzed the data, performed the computation work, authored or reviewed drafts of the paper, and approved the final draft.
- Ming Jiang performed the experiments, prepared figures and/or tables, authored or reviewed drafts of the paper, and approved the final draft.

### Data Availability

The code is available at GitHub: https://github.com/lpfworld/DMPNet.
The ShanghaiTech Part A and Part B is available at Kaggle: https://www.kaggle.com/tthien/shanghaitech.
The UCF-QNRF is available at: https://www.crcv.ucf.edu/data/ucf-qnrf/
The UCF_CC_50 is available at: https://www.crcv.ucf.edu/data/ucf-cc-50/.

### Supplemental Information

Supplemental information for this article can be found online at http://dx.doi.org/10.7717/peerj-cs.902#supplemental-information.

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
