# Peer review of "DMPNet: densely connected multi-scale pyramid networks for crowd counting"

_PeerJ Computer Science, doi:10.7717/peerj-cs.902_

## Round 0.1 · original submission · Major Revisions

My suggestions and comments to the authors are as follows:
1) the title could be modified to "DMPNet: Densely connected multi-scale pyramid networks for crowd counting"
2) The abstract should begin with the problem statement, rather than directly talking about the performance.
3) Legends of Fig. 2, 3 and 4 need more explanation.
4) In the conclusion section, mention the limitation of the proposed model, and future work.

·

Basic reporting

The paper, at a high level, presents a way to cope with the scale variation problem in Crowd Counting research. The authors clearly write about their motivation for coming up with the proposals and also compare the counting performance with state-of-the-art methods on 4 major datasets. The main contributions of the paper are the proposed Multi-scale Pyramid Networks (MPN), which are densely connected for better information retention throughout the deep networks. I shall comment on the basic components of the paper that could be improved as follows:

1. I believe that the very first paper that uses dense connections is “Dense Scale Network for Crowd Counting” (DSNet in Table 2) [1]. Thus, (Dai et. al., 2021) should be mentioned when introducing the dense connections.

2. The published year of DSNet is not consistent. (Dai et. al., 2019) or (Dai et. al., 2021)?
In crowd counting research, to “keep the input and output resolutions unchanged”, the encoder-decoder structure (See [3] and [5]) is usually utilized. The introduction to the structure is worth adding for better context.

3. In the RELATED WORK Section, to better support the difference between this work and others, there should be examples of previous papers that add “extra perspective maps or attention maps”, for instance [2],[3], and [4]. Specifically [3] also adopts multiple branches of convolution with different kernel sizes, namely the ASPP structure.

4. In Figure 4., please carefully check that #groups = 2 or 4? Based on Figure 4., the incoming feature channels are divided into 2 groups?

5. About describing the configurations (kernel_size, #output_channel, ...) of a convolution operation, It is preferable to use \times over the “*” notation. The “*” notation may be reserved for the convolution operation itself.

6. In Figure 5, Is there any particular reason, the larger kernel_size is, the smaller output channels and the groups are? Is this just to balance computational resources between branches?

7. (Typo) Change “MFFE” to “MFFN”?

8. (Typo) What is G (with \theta parameterization) in the density level consistency loss? Should G be F?

9. Is C in the calculation of MAE loss, the summation over all pixels? The definition of C can be made more clear. Do I and D^{GT} notations have the same meaning? As I_{i} cannot appear twice, please revise the MAE loss formulation.

10. \alpha and \beta for weighting the losses are chosen based on what criterion? Experiences? Random? or validation count performance?

References
[1] Dai, Feng, et al. "Dense scale network for crowd counting." Proceedings of the 2021 International Conference on Multimedia Retrieval. 2021.
[2] Zhu, Liang, et al. "Dual path multi-scale fusion networks with attention for crowd counting." arXiv preprint arXiv:1902.01115 (2019).
[3] Thanasutives, Pongpisit, et al. "Encoder-Decoder Based Convolutional Neural Networks with Multi-Scale-Aware Modules for Crowd Counting." 2020 25th International Conference on Pattern Recognition (ICPR). IEEE, 2021.
[4] Shi, Miaojing, et al. "Revisiting perspective information for efficient crowd counting." Proceedings of the IEEE/CVF Conference on Computer Vision and Pattern Recognition. 2019.
[5] Jiang, Xiaolong, et al. "Crowd counting and density estimation by trellis encoder-decoder networks." Proceedings of the IEEE/CVF Conference on Computer Vision and Pattern Recognition. 2019.
[6] Chen, Liang-Chieh, et al. "Deeplab: Semantic image segmentation with deep convolutional nets, atrous convolution, and fully connected crfs." IEEE transactions on pattern analysis and machine intelligence 40.4 (2017): 834-848.

Experimental design

1. On defining the evaluation metrics, you can use D^{GT} instead of Z`. The notations need consistency.
There are no “bolded” numbers, indicating the best results in any Table.

2. Current SOTA methods are incomplete, [2], [3], [4], and [5] are missing.

3. In Table 3., it is not clear whether (w/ LPN, w/o GPN) and (w/ GPN, w/o LPN) are trained with MFFN or not? In the 3rd row, MPN should be MFFN? Another typo?

4. From Table 4., it is still unclear how many blocks of MPN blocks should be densely connected? In this paper, It is 3, but how about 1, 2, or even 4? This may be addressed in the ablation study section.

5. This paper recommends the use of the group convolution (over the normal one) but does not compare the number of trainable parameters of the proposed Multi-scale Pyramid Network, when G varies and the case that G = 1 for all the branches.

References
[1] Dai, Feng, et al. "Dense scale network for crowd counting." Proceedings of the 2021 International Conference on Multimedia Retrieval. 2021.
[2] Zhu, Liang, et al. "Dual path multi-scale fusion networks with attention for crowd counting." arXiv preprint arXiv:1902.01115 (2019).
[3] Thanasutives, Pongpisit, et al. "Encoder-Decoder Based Convolutional Neural Networks with Multi-Scale-Aware Modules for Crowd Counting." 2020 25th International Conference on Pattern Recognition (ICPR). IEEE, 2021.
[4] Shi, Miaojing, et al. "Revisiting perspective information for efficient crowd counting." Proceedings of the IEEE/CVF Conference on Computer Vision and Pattern Recognition. 2019.
[5] Jiang, Xiaolong, et al. "Crowd counting and density estimation by trellis encoder-decoder networks." Proceedings of the IEEE/CVF Conference on Computer Vision and Pattern Recognition. 2019.
[6] Chen, Liang-Chieh, et al. "Deeplab: Semantic image segmentation with deep convolutional nets, atrous convolution, and fully connected crfs." IEEE transactions on pattern analysis and machine intelligence 40.4 (2017): 834-848.

Validity of the findings

1. (Important) Since this paper would be perceived (if it gets published) as an incrementally improved version of DSNet, some audiences might get confused by the fact that the DMPNet only outperforms DSNet in terms of lowering MSE computed from the ShanghaiTech Part_B dataset. Hence, I advise that you can also compare the number of trainable parameters or inference speed for better justification of your proposed DMPNet. I guess that DMPNet might contain fewer parameters as you employed the group convolution operation.

2. As I mentioned above, it is hard to justify that DMPNet has “better in both mean absolute error (MAE) and mean squared error (MSE)” at the current state of the paper. So, there may be doubts whether “MPN can effectively extract multi-scale features” or not?

3. On lines 290-291, the claim that “Our DMPNet has well solved the problems of crowd occlusion, perspective distortion, and scale variations” is a bit exaggerated. It is unclear how exactly you deal with the perspective distortion? Is this solely based on your experimental results of the UCF_QNRF dataset?

Reviewer 2 ·

Basic reporting

The authors proposed a Densely connected Multi-scale Pyramid Network (DMPNet) based on VGG and Multi-scale Pyramid Network (MPN). The latter consists of three modules that aim at extracting features at different scales. The authors compared performances obtained by DMPNet with other methods at state of the art.

1. Figures and Table should be placed before the bibliography
2 Related work section should include all methods used for comparison.
3. Introduce every acronym before using it (e.g., CNN, ASNet, etc.)
4. Viresh etal. (line 97-98) and CP-CNN method (Table 2) are not present in the bibliography
5. Figures 3 and 4 have the same caption.
6. Captions should be improved
7. In table 2, no results are highlighted in bold
8. Figure 1 presents "Table 1" in the caption
9. In the manuscript and Figure 5, the symbol "*" was used. It is necessary to replace it with "x" (in Latex, \times).
10. lines 35-36, 53-54, 103-106, 107-108, 195-196 need references

Experimental design

The method is described with sufficient details. Moreover, the authors provide the code.

1. The training methods section can be integrated into Experimental and Discussion
2. The measure MSE (Section "Evaluation Metrics") is the "Root Mean Squared Error" (RMSE)
3. lines 267-271, 280-282 can be improved

Validity of the findings

1. The ablation experiment section can be improved by studying these effects on the other considered data sets.

Additional comments

1. There are some typos (e.g., SOAT --> SOTA)

---

## Round 0.2 · accepted · Accept

The manuscript has been revised as per the reviewer's comments. Your manuscript may be accepted for publication.

Reviewer 2 ·

Basic reporting

no comment

Experimental design

no comment

Validity of the findings

no comment